# Development of a Mobile Sensory Device to Trace Treatment Conditions for Various Medical Plasma Source Devices

**DOI:** 10.3390/s22197242

**Published:** 2022-09-24

**Authors:** Ihda Chaerony Siffa, Torsten Gerling, Kai Masur, Christian Eschenburg, Frank Starkowski, Steffen Emmert

**Affiliations:** 1ZIK Plasmatis, Leibniz Institute for Plasma Science and Technology (INP Greifswald), Felix-Hausdorff-Straße 2, 17489 Greifswald, Germany; 2Diabetes Competence Centre Karlsburg (KDK), Greifswalder Straße 11, 17495 Karlsburg, Germany; 3Orthopädie-Technik-Service Aktiv GmbH, Gützkower Landstraße 36–40, 17489 Greifswald, Germany; 4Clinic and Policlinic for Dermatology and Venerology, University Medical Center Rostock, Strempelstraße 13, 18057 Rostock, Germany

**Keywords:** plasma medicine, medical plasma sources, treatment documentation, FFT-based analysis, sound detection, device development

## Abstract

The emerging use of low-temperature plasma in medicine, especially in wound treatment, calls for a better way of documenting the treatment parameters. This paper describes the development of a mobile sensory device (referred to as MSD) that can be used during the treatment to ease the documentation of important parameters in a streamlined process. These parameters include the patient’s general information, plasma source device used in the treatment, plasma treatment time, ambient humidity and temperature. MSD was developed as a standalone Raspberry Pi-based version and attachable module version for laptops and tablets. Both versions feature a user-friendly GUI, temperature–humidity sensor, microphone, treatment report generation and export. For the logging of plasma treatment time, a sound-based plasma detection system was developed, initially for three medically certified plasma source devices: kINPen^®^ MED, plasma care^®^, and PlasmaDerm^®^ Flex. Experimental validation of the developed detection system shows accurate and reliable detection is achievable at 5 cm measurement distance in quiet and noisy environments for all devices. All in all, the developed tool is a first step to a more automated, integrated, and streamlined approach of plasma treatment documentation that can help prevent user variability.

## 1. Introduction

### 1.1. Plasma Medicine

Plasma, also known as the fourth state of matter, has been a subject of continuous research and applied in many different fields [1]. Recent advancement in low-temperature plasma physics research has shown potential applications of plasma in the field of medicine [2]. The primary application of plasma in medicine predominantly uses non-thermal plasma at atmospheric conditions also known as cold atmospheric pressure plasma (CAP).

Non-thermal plasma such as CAP can be generated by exerting electrical energy to a feed gas. Most of the applied energy is in the form of highly energetic electrons. These electrons ionize the feed gas molecules such as argon, helium, oxygen, nitrogen, or a mixture of such gases as found in the ambient air. The ionized gas molecules remain in a low-energetic state due to being heavier than the electrons, thus allowing the overall temperature of the plasma to be relatively low, typically below 40 ${}^{\circ }$C. The ionization processes of the feed gas give rise to the emission of electromagnetic radiations of several forms, e.g., ultraviolet (UV), and infrared or heat [3,4]. The low temperature of CAP is desirable for biomedical applications, especially when a direct application of the plasma on the surface of a biological tissue is necessary.

The application of CAP has shown to be effective in several therapeutic settings, such as skin disinfection and the treatment of chronic wounds [4,5,6]. The efficacy of CAP in these therapeutic settings may be attributed to the production of chemical species such as reactive oxygen species (ROS) and reactive nitrogen species (RNS) in combination with UV radiation, visible light, and mild heat, which promote wound healing and other medical effects such as anti-bacterial, anti-fungal, anti-itch, anti-pain, anti-inflammatory and anti-scarring as well as anti-tumoral effects [2,3,7,8,9,10].

### 1.2. Medical Plasma Source Devices

The application of CAP is achieved with plasma source devices, and in the field of plasma medicine, atmospheric pressure plasma jet (APPJ) and dielectric barrier discharge (DBD) are the most commonly used plasma source types [11,12]. As briefly mentioned in Section 1.1, to generate the plasma, the electrical energy (usually in form of high voltage) has to be exerted to a feed gas. The applied high voltage generates a locally amplified electrical field and ionizes the feed gas until a plasma discharge occurs [13].

Medical APPJ plasma sources have a cylindrical structure equipped with an exit nozzle usually configured with one or two electrodes inside [14]. A steady flow of gas is fed into this structure, where it becomes partially ionized generating a plasma discharge, which is streaming outside the nozzle such as a jet, which is also called an effluent [3,15,16]. This plasma effluent is then applied to the surface of a target. For example, in medical applications, the target would be biological tissues such as wounds and skins. To achieve a steady flow of the feed gas, a pressurized gas tank and a flow controller are usually needed.

As for medical DBD plasma sources, a dielectric isolated high-voltage electrode is placed directly a few millimeters above the target tissue, which acts as a ground electrode. The plasma is generated in this gap, causing a direct application of the plasma on the tissues [13]. For DBDs, ambient atmospheric air is mostly used as the feed gas.

Many years of plasma source developments have given birth to several plasma source devices that are medical-application ready. For example, an APPJ device such as kINPen^®^ MED (Leibniz Institute for Plasma Science and Technology/neoplas med GmbH, Greifswald, Germany), and DBD devices such as plasma care^®^ (terraplasma medical GmbH, Garching, Germany) and PlasmaDerm^®^ Flex (CINOGY Technologies GmbH, Duderstadt, Germany) [17] have been CE-certified [5].

Another technique that also helps these devices achieve low-temperature plasma generation is the non-continuous excitation of the feed gas. The excitation process is pulsed (on/off pulse) at a certain frequency, depending on the device [4,10].

### 1.3. Chronic Wound Treatment Using Plasma Sources

Wounds come in various shapes and sizes on different parts of the body. Due to different characteristics of the plasma of APPJ sources and DBD sources on the tissue, one source may be more suitable than the other to be used on a certain wound condition or shape.

Chronic wound treatment with CAP is an emerging topic that is not a decade old and still lacks stringent guidelines and procedures [12,18]. Clinicians are, however, continuously working on perfecting this treatment procedure as this treatment is gaining more traction in recent years. In the study by Brehmer et al., (2015), the PlasmaDerm^®^VU-2010 device was used in their treatment protocol. The plasma was applied for 45 s per cm^2^ of the wound surface. Depending on the area of the wound, a treatment can take around 11 min for each treatment session. The treatment was performed three times a week for the course of eight weeks [19]. In another study carried out by Ulrich et al. (2015) using the kINPen^®^ MED device, the plasma was applied on the wounds at a distance of 7–8 mm from the source nozzle, roughly for one minute per cm^2^ of the wound surface. The treatment was completed three times a week for the course of three weeks [20].

Apart from the duration of plasma exposure on the target, humidity is also suspected to affect the outcome of the treatment—especially for DBD plasma sources, working with ambient air. It was shown for the kINPen device that the (artificially added) humidity of the feed gas and ambient air strongly influence the excited plasma species [21,22], which may affect the stability and reproducibility of studies in plasma medicine.

Documentation is also a substantial part in wound treatment with which the healing progress can be assessed and monitored. This includes the general information of the patient, the photograph of the wound, the wound assessment of each treatment (e.g., wound size), and other parameters. In the plasma treatment, ideally, one may also need documenting which plasma source device is used, how long the plasma is applied, and whether it was applied correctly [12].

### 1.4. Present Work

This paper presents the development of an initial prototype of a mobile sensory device, which is named MSD. This mobile device is intended to be used along with the previously mentioned plasma source devices in plasma-based wound treatment, especially in the ambulant settings (here, the most typical ambulance setting is a plasma wound treatment at the patient’s place or residential buildings). The main function of MSD is to collect and document necessary information, which can be used to better aid clinicians and researchers in assessing the efficacy of the treatment. Currently, the developed MSD prototype provides the following features:Automatic documentation of plasma treatment duration—precise detection and logging of the plasma usage for the kINPen^®^ MED, plasma care^®^, and PlasmaDerm^®^ Flex devices in comparison to manual detection and tracking to prevent user variability;Logging of ambient conditions—humidity and temperature;Graphical User Interface (GUI) for user-friendly operation;Plasma treatment report generation and export;Compact design for high mobility—as a module attachment for tablets or laptops, or as a standalone portable device based on a Raspberry Pi single-board computer.

## 2. Materials and Methods

### 2.1. Plasma Source Sound Analysis

The ionization process in plasma generation results in several forms of electromagnetic emissions. At first glance, detecting the presence of plasma may be intuitively done visually by detecting the electromagnetic emissions in the visible spectrum. However, visual detection requires a camera system that must face the emission source unobstructed at all times to achieve real-time detection and tracking. Due to the way the plasma source devices are used and treatment distance of down to 1 mm scale as well as the average size of a camera system, the use of such a system is not practicable for this task. There is, however, another emission resulting from the plasma generation process, which is sound emission. This opens up the possibility of developing a plasma detection system based on the generated sound frequency signature of each device. A sound frequency-based plasma detection system is considered more practicable because of two things: the sounds produced by the devices have enough energy (the loudness of each device’s sound signal was measured using a sound-level meter (SLM) smartphone application (https://play.google.com/store/apps/details?id=com.gamebasic.decibel, accessed on 12 May 2021) installed on a Samsung Galaxy J7 Prime. The signals were measured at a ±10 cm distance from the source for 30 s. The kINPen^®^ MED produces a sound with 55.56 dB loudness, PlasmaDerm^®^ Flex 49.1 dB, and plasma care^®^ 22.1 dB) to propagate throughout the environment, which allows a less constrained placement of the sound sensor; and it is more feasible to construct a small sound detection system compared to a camera-based one.

The mentioned plasma source devices generate sound emissions within the audible frequency range (20 Hz–20 kHz) as well as above. An electro-acoustic investigation conducted by Law et al., (2014) on the kINPen^®^ MED device showed that the plasma generation pulse frequency of 2.5 kHz with a 50% duty cycle correlates with a sound generation of the same frequency and its harmonics [23]. This demonstrates that the plasma generation pulse frequency used in a plasma source device directly affects the generated sound signature. This sound signature is a result of the repetitive discharge events in the immediate ambient air and feed gas during the plasma generation process.

For further observations, the sound signals from each device were independently recorded using smartphones at a distance of 10 cm for 10 s, with a sampling frequency of 44,100 Hz, in a quiet confined room (20 dB of ambient noise). The recordings were converted to WAV files with 16-bit signed integer data type retaining the sampling frequency, and the sound spectra were analyzed by means of Fourier transform. The fast Fourier transform (FFT) algorithm with Hamming (smoothing) window and Z-frequency-weighting were used for this analysis, which was implemented and visualized with the Python programming language (v3.7.3) utilizing the SciPy (v1.4.1), Matplotlib (v3.2.1), and NumPy (v1.18.1) libraries [24,25,26,27]. Figure 1a shows the frequency spectrum of the kINPen^®^ MED device sound signals observed up to 20 kHz presented in decibels (dB) relative to full scale. Several of the highest peaks can be seen occurring at 2.55 kHz and its harmonics, which is in accordance with [23]. The peak at 17.8 kHz has the highest magnitude (−33.8 dB), and the lowest magnitude (−68.9 dB) is observed at 5.01 kHz.

The plasma care^®^ sound frequency spectrum, presented in Figure 1b, shows four highest peaks occurring at 4.04 kHz (−21.8 dB), 8.08 kHz (−22.9 dB), 12.12 kHz (−38.7 dB), and 16.16 kHz (−41.2 dB). An increment of around 4 kHz can be inferred from these frequencies, indicating the use of 4.04 kHz pulsing frequency in the device, where the rest of the frequencies act as the harmonics.

Similarly, the frequency spectrum of the PlasmaDerm^®^ Flex device sound signals is presented in Figure 1c. Here, there are significantly more prominent peaks occurring. If most peaks are assumed to be the harmonics, then one thing that can be deduced is that the pulsing frequency is lower than the previous devices. A closer look at the frequency spectrum around the highest peak (4.77 kHz) shows that several other prominent peaks can be inferred at: 4.77 kHz (−36.3 dB), 4.47 kHz (−36.9 dB), 3.87 kHz (−38.9 dB), 4.17 kHz (−39.7 dB), and 5.07 kHz (−45.5 dB). These frequencies suggest a difference of around 300 Hz between the peaks, which indicates the use of pulsing frequency of around 300 Hz for this plasma source device [11].

The spectrogram of each device’s sound signals was also computed to see the consistency of the frequency signatures. As shown in Figure 2, the frequency signatures appear to be consistent at FFT size of 100 ms observed up to 10 s.

### 2.2. Detection Strategy

Distinct frequency peaks of the sound signals produced by the devices and the consistency of the sound frequency signatures at a considerably small FFT size (100 ms) allow a straightforward detection system based on each device’s sound frequency signature. The basic idea is to record the sound signals periodically and compute the FFT, then check if the magnitudes of the pulsing frequency (now denoted as a fundamental frequency, $f_{f}$) and its harmonics are above some threshold values.

As a first step, the device sound signals, $s=\left(s_{0}\;s_{1}\;s_{2}...\right)$, are recorded with a sampling frequency rate, $f_{s}$, of 44,100 Hz for a certain $t^{\mathrm{rec}}$. The signals are then processed to obtain the real-valued/absolute FFT values, which is denoted as an array $S=\left(S_{0}\;S_{1}\;S_{2}...S_{n-1}\right)$ with the length of *n*. Due to how the FFT is computed, the value of *n* depends on $f_{s}$ and $t^{\mathrm{rec}}$, i.e., how many data points are sampled within a certain amount of time. The audible frequency is defined as an array $f=\left(f_{0}\;f_{1}\;f_{2}...f_{n-1}\right)=\left[0...(f_{s}/2)-1\right]$. To ease the array indexing for the detection process later on, having S with the same length as f is necessary. This results in $f_{s}/2=22,050\;\mathrm{Hz}$ following the Nyquist–Shannon sampling theorem such that the index of each element in S corresponds to the exact frequency value within the audible frequency range $\left(S_{k}\;|\;k\in f\right)$. To achieve this, a $t^{\mathrm{rec}}$ of one second is needed, following
(1)$$n=\frac{t^{\mathrm{rec}}\cdot f_{s}}{2}.$$

However, $t^{\mathrm{rec}}=1$ s implies that the time required to complete one detection instance, $t^{\det}$, is greater than one second, which is sub-optimal for this task. For optimization, $t^{\mathrm{rec}}=0.1$ s is used; then, ***S*** is resized to have the length of *n* = 22,050 using a linear interpolation-based scaling method. At this point, a resized frequency spectrum array, $S^{\mathrm{res}}=\left(S_{0}^{\mathrm{res}}\;S_{1}^{\mathrm{res}}\;S_{2}^{\mathrm{res}}...S_{n-1}^{\mathrm{res}}\right)$, is obtained.

The detection threshold values are computed in each detection instance. The values can be obtained by first fitting a high-order polynomial curve to $S^{\mathrm{res}}$ in the form of decibel full scale (dBFS), which is denoted as $A=\left(A_{0}\;A_{1}\;A_{2}\;...\;A_{n-1}\right)$. The *j*-th order polynomial function, $p\left(x\right)=p_{0}+p_{1}x+p_{2}x^{2}+...+p_{j}x^{j}$, can be obtained by solving the unknown parameters (coefficients), *p*, of the following system of linear equations:(2)$$\left[\begin{array}{c} A_{0}\\ A_{1}\\ A_{2}\\ \vdots \\ A_{n-1} \end{array}\right]=\left[\begin{array}{ccccc} 1 & f_{0} & f_{0}^{2} & \cdots & f_{0}^{j}\\ 1 & f_{1} & f_{1}^{2} & \cdots & f_{1}^{j}\\ 1 & f_{2} & f_{2}^{2} & \cdots & f_{2}^{j}\\ \vdots & \vdots & \vdots & \ddots & \vdots \\ 1 & f_{n-1} & f_{n-1}^{2} & \cdots & f_{n-1}^{j} \end{array}\right]\cdot \left[\begin{array}{c} p_{0}\\ p_{1}\\ p_{2}\\ \vdots \\ p_{j} \end{array}\right].$$

Now, the *k*-th threshold value, $l_{k}$, can be obtained by adding a predefined adjustable term into the polynomial function:(3)$$l_{k}=\left(p_{0}+p_{1}k+p_{2}k^{2}+\cdots +p_{j}k^{j}\right)+\sigma \eta ,$$
where σ is the standard deviation of A given by
(4)$$\sigma =\sqrt{\frac{\sum_{k=0}^{n-1}{\left(A_{k}-\mu \right)}^{2}}{n}},$$
with the mean,
(5)$$\mu =\frac{1}{n}\sum_{k=0}^{n-1}A_{k},$$
and η is a device-specific adjustable factor. Finally, an array of threshold values for A, $l=\left(l_{0}\;l_{1}\;l_{2}\;...\;l_{n-1}\right)$, is obtained.

A plasma is detected when a certain amount of the indexed elements in A are above the threshold values of the same indices. The indices for these elements are a set of target frequencies, $f^{\mathrm{target}}\subset f$ and $f_{f}\in f^{\mathrm{target}}$, which is specific to each device. For example, $f^{\mathrm{target}}=\left\{4040,\;8080,\;12,120,\;16,160\right\}$ can be used for the plasma care^®^ device based on the sound analysis, which is basically the plasma care^®^ device’s $f_{f}$ and its harmonics. For target frequencies of other devices, see Table 1.

Using the polynomial regression method as the base of the threshold curve creates the thresholding process that is adaptive and specific to each target frequency. Furthermore, the last term of Equation (Equation 3) provides the adjustability of the adaptive thresholding, which directly influences the sensitivity and the specificity of the detection. Figure 3 shows the plot of the 12th order polynomial threshold curves of the PlasmaDerm^®^ Flex sound frequency spectrum with different η values on top of the PlasmaDerm^®^ Flex frequency spectrum. Based on Figure 3, it is clear that finding an optimal value of η is crucial for a robust detection system given changing ambient conditions under ambulant or mobile usage.

In the next step, a feedback form frequency-domain comb filter is proposed to be used before the detection process to attenuate unwanted frequencies other than $f_{f}$ and its harmonics, which further increases the robustness of the detection system. The filter function is given by
(6)$$H\left[k\right]=\frac{1}{\sqrt{\left(1+\alpha^{2}\right)-2\alpha \;\cos\left(2\pi k\xi \right)}},$$
where $\xi \left(f_{f}\right)=\frac{1}{f_{f}}-1$ represents the spacing frequency between notches, and α represents the steepness of the filtering. Evaluating Equation (Equation 6) for every k∈f gives the filter array $H=\left(H_{0}\;H_{1}\;H_{2}\;...\;H_{n-1}\right)$ that has the same length as $S^{\mathrm{res}}$.

Finally, the filtered sound frequency spectrum, $S^{\mathrm{filtered}}=\left(S_{0}^{\mathrm{filtered}}\;S_{1}^{\mathrm{filtered}}\;S_{2}^{\mathrm{filtered}}\;...\;S_{n-1}^{\mathrm{filtered}}\right)$, is the Hadamard product of $S^{\mathrm{res}}$ and the normalized filter array, $\hat{H}=\left({\hat{H}}_{0}\;{\hat{H}}_{1}\;{\hat{H}}_{2}\;...\;{\hat{H}}_{n-1}\right)$, which can be written as $S^{\mathrm{filtered}}=S^{\mathrm{res}}⊙\hat{H}$. In this case, H is normalized by its maximum value,
(7)$${\hat{H}}_{k}=\frac{H_{k}}{\max(H)},\;\;\;\;\;k\in f,$$
where ${\hat{H}}_{k}$ is the *k*-th value of $\hat{H}$, and $H_{k}$ is the *k*-th value of H. Note that the Hadamard operation between $S^{\mathrm{res}}$ and $\hat{H}$ is only computable when both arrays have the same length, which is always the case in this filtering step. Figure 4 shows $\hat{H}$ with different α values and $f_{f}=2550$ Hz.

By this filtering method, frequencies other than $f^{\mathrm{target}}$ are attenuated, thus making $f^{\mathrm{target}}$ more pronounced. An example of $S^{\mathrm{filtered}}$ of kINPen^®^ MED sound signals, with a filter configuration α=0.99, and $f_{f}=2550$ Hz as well as the corresponding threshold curve is shown in Figure 5.

Using this adaptive threshold curve method always requires special consideration: that is, when the system only listens to white noise. Due to the randomness of white noise, the calculated frequency spectrum may have similar characteristics to the target frequency spectrum, which increases the risk of false-positive detection. To mitigate this problem, the detection system first must know when the sound signals that come into it are low, i.e., no plasma sounds are present and the environment is quiet. If the signals are low, different α and η values must be used to adjust the responsiveness of the detection. A combination of large α and η values will result in a low detection responsiveness and vice versa. In this detection strategy, dynamic α and η values are used, which change depending on the magnitude of the incoming signals. Since α and η should not be chosen arbitrarily, the system must choose these values within certain ranges, $\left[\alpha^{\min},\alpha^{\max}\right]$ and $\left[\eta^{\min},\eta^{\max}\right]$. Finally, one may calculate the mean of $s^{\mathrm{res}}$ to determine the loudness of the incoming signals, but before that, the upper and lower limit of the magnitude, $\left[b^{\min},b^{\max}\right]$, must be defined. Here, all minimum and maximum values are obtained empirically. This quietness check step is completed right before the filtering and threshold generation steps. This step is not only beneficial in a quiet case, as it may also increase the detection system robustness in the case of detecting plasma sounds in a loud environment. Note that this quietness check is completed in every detection instance.

Moving on to the detection step, $S^{\mathrm{filtered}}$ is converted to dB relative to full scale or dBFS, resulting in $A^{\mathrm{filtered}}=\left(A_{0}^{\mathrm{filtered}}\;A_{1}^{\mathrm{filtered}}\;A_{2}^{\mathrm{filtered}}\;...\;A_{n-1}^{\mathrm{filtered}}\right)$. Each *i*-th target frequency, $f_{i}^{\mathrm{target}}$, is detected when the magnitude of $A^{\mathrm{filtered}}$ at the index of $f_{i}^{\mathrm{target}}$ is greater than the value of l at the same index value. However, since $f^{\mathrm{target}}$ depends on an external plasma source device calibration, a detection margin is used. Let c be an empty array where each detected frequency peak within one detection instance is stored. The *i*-th frequency detection can be written as
(8)$$c_{i}=\left\{\begin{array}{cc} 1 & \mathrm{if}\;\;\;A_{m}^{\mathrm{filtered}}>l_{m}\\ 0 & \mathrm{otherwise} \end{array}\right.,$$
with margin index, m∈ν, where
(9)$$\nu =\underset{k\in m}{arg max}A^{\mathrm{filtered}}\left[k\right],$$
and $m=\left[f_{i}^{\mathrm{target}}-\epsilon ,f_{i}^{\mathrm{target}}+\epsilon \right]\cap f$ is an array consisting of margin indices for a certain $f_{i}^{\mathrm{target}}$, and $0\geq \epsilon <<f_{f}$ is a predefined device-specific margin value. The final detection state, $c^{\mathrm{final}}$, is defined as
(10)$$c^{\mathrm{final}}=\left\{\begin{array}{cc} 1 & \mathrm{if}\;\;\;\sum c\geq \theta \\ 0 & \mathrm{otherwise} \end{array}\right.,$$
where θ is a device-specific value that tells how many peaks must be detected for one detection instance to be considered valid, which must satisfy $\theta \leq \frac{n}{f_{f}}$.

Finally, $t^{\det}$ is the sum of $t^{\mathrm{rec}}$ and the elapsed processing time, $t^{\mathrm{proc}}$. $t^{\det}$ of each detected case ($c^{\mathrm{final}}=1$) is stored in the array z, and the length of the plasma treatment time, $t^{\mathrm{plasma}}$, at the end of the treatment is the sum of all elements in z. The computation steps of this sound detection and plasma treatment time measurement system are summarized in Algorithm 1.

This detection strategy is implemented and visualized with the Python programming language (v3.7.3) utilizing the SciPy (v1.4.1), Matplotlib (v3.2.1), NumPy (v1.18.1), and PyAudio (v0.2.11) libraries [24,25,26,27].  
**Algorithm 1:** Plasma sound detection and treatment time measurement.**Data**: $t^{\mathrm{rec}},f_{f},\alpha ,b,\eta ,\epsilon ,\theta$**Result**: $t^{\mathrm{plasma}}$Listen←Truez←EmptyArray()**while**Listen=True**do**
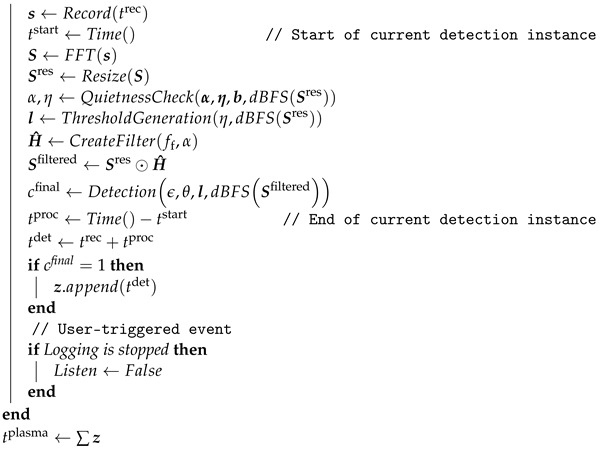


### 2.3. Hardware and GUI Overview

High mobility is part of the core requirement of the MSD’s development, as plasma wound treatment is becoming more relevant for ambulant patients [28]. To fulfill this requirement, MSD was developed with two different hardware versions in mind: as a standalone device and as an attachable module for tablets or laptops.

For the standalone version of MSD, a single-board computer (Raspberry Pi 3 Model B, Raspberry Pi Ltd., Cambridge, UK) with a Linux-based Raspbian Buster 10 operating system (OS) is used as the main processing unit. A condenser microphone with a frequency response within the audible range is used to capture the sound signals generated by the plasma source. The microphone is intended to be attached to the plasma sources or placed close to it for an optimal reading/recording of the sound signals. An RS PRO Lavalier wired microphone 1 kΩ (RS Components GmbH, Frankfurt am Main, Germany) is selected for this recording purpose having a frequency response of 30 Hz to 18 kHz. The microphone is connected to the Raspberry Pi via a USB audio card (StarTech ICUSBAUDIOB, StarTech.com Ltd., London, ON, Canada). For the measurements of ambient temperature and humidity, a DHT22 sensor (AM2302 Waveshare, Waveshare Electronics, Shenzhen, China) is used, which is interfaced with the Raspberry Pi. DHT22 is a commonly used sensor in the prototyping phase of IoT (Internet of Things) system developments [29], which is capable of performing periodic measurements every around two seconds, which is adequate for the task.

The operation of MSD requires an input–output (i/o) system. For this part, a 5-inch HDMI touchscreen display module (Waveshare Electronics, Shenzhen, China) is used, which enables a touchscreen-based i/o system keeping the design as compact as possible. To further increase the mobility, the standalone MSD is equipped with a battery module (ISY IPP 5000-SL-BK Powerbank 5000 mAh, Imtron GmbH, Ingolstadt, Germany). The correctness of the documentation date and time are essential, and to ensure this, the system utilizes a real-time clock (RTC) module (Seeed Pi RTC, Seeed Technology Inc., Shenzhen, China).

The device casing for the standalone MSD was designed and 3D-printed. The final assembly of the standalone MSD, shown in Figure 6a, has a dimension of 125×190×46 mm${}^{3}$, and a microphone wire length of 800 mm.

As for the attachable module, a Windows or Linux-based tablet/laptop can be used as the main processing unit, and in this development case, an HP^®^ Pavilion x360 model 15-dq1450ng laptop (HP Deutschland GmbH, Ratingen, Germany) with the Windows 10 OS is used. A microphone with the same specification as in the standalone MSD can be used, which can be directly plugged into the laptop without requiring any USB audio card. For the DHT22 sensor implementation, a microcontroller is required to interface the DHT22 sensor with the laptop. The Arduino Nano microcontroller (Arduino S.r.l., Monza, Italy) is used for this interface, which is connected directly to the laptop via a USB port. The laptop already has an in-built RTC module, which is commonly found in modern laptops. Similarly, a casing was designed and 3D-printed for the microcontroller and the sensor, with the resulting dimension of 72×72×32 mm^3^. The assembled attachable MSD with a laptop is shown in Figure 6b.

Lastly, for a good user experience when operating MSD, a GUI system was developed using Python (v3.7.3) and the Kivy (v1.11.1) framework. The GUI consists of several different screens, which are displayed according to the current functionality and process. The user navigates through these different screens by pressing the provided virtual buttons on the screens. Furthermore, the GUI is available in the English and German languages. Figure 7 shows the overview of the latest iteration of the GUI (software version 0.3.0) taken from the attachable module version installed on the windows 10 OS, and Figure 8 illustrates the working principle of MSD used along with the kINPen^®^ MED device.

## 3. Results and Discussion

### 3.1. Experimental Validation of the Plasma Sound Detection System

MSD was developed in two versions: standalone version and attachable module version. To validate the developed plasma sound detection system, the standalone version of MSD was used and tested on all three plasma source devices. The MSD’s microphone was placed at different measurement distances (5 cm, 10 cm, 15 cm, 20 cm, 25 cm, 30 cm, 1 and 2 m) from the plasma sound source in four different measurement conditions: (1a) plasma sound in a quiet room (ambient condition), (1b) quiet room without plasma source and any artificial noise, (2a) plasma source and rock music noise, (2b) only rock music noise, (3a) plasma sound and newscast noise, and (3b) only newscast noise. A smartphone was used to generate the rock music (Nirvana, “Smells Like Teen Spirit” (https://www.youtube.com/watch?v=hTWKbfoikeg, from 0:45 to 1:25, accessed on 5 July 2021) with 61 dB loudness at 10 cm measurement distance) [30] and radio news (Brexit: UK leaves the European Union, BBC News (https://www.youtube.com/watch?v=iBRcg05rzHs, from 0:00 to 0:30, accessed on 5 July 2021) with 47 dB loudness at 10 cm measurement distance) noises. The smartphone was placed one meter away from the plasma source.

For the measurements using the medical DBD plasma source devices, plasma care^®^ and PlasmaDerm^®^ Flex, human skin was used as the target. In the case of PlasmaDerm^®^, a close proximity to a target is always required for plasma generation and thus sound generation. In regard to plasma care^®^, the device generates sound whether a target is present or not. However, when the device is applied on a target following the suggested application procedure including the use of a spacer, the generated sound becomes quieter due to the device–target interface. As for the kINPen^®^ MED device, there is no humanly noticeable difference when using the device with or without a target.

The measurement time, $t^{\mathrm{measure}}$, was set to 30 s and automatically controlled by a measurement script in MSD to prevent user variability. The measurement script collects every $c^{\mathrm{final}}$ within $t^{\mathrm{measure}}$, and from the collected data, the sensitivity and specificity of the detection system for each device can be calculated, following
(11)$$\mathrm{TPR}=\frac{\mathrm{TP}}{\mathrm{TP}\;+\;\mathrm{FN}},$$
and
(12)$$\mathrm{TNR}=\frac{\mathrm{TN}}{\mathrm{TN}\;+\;\mathrm{FP}},$$
where TPR (True Positive Rate) is the sensitivity, and TNR (True Negative Rate) is the specificity of the detection system. TP (True Positive) is a measure of how many times MSD *correctly* detects the plasma sound when the plasma source device is active at each measurement distance throughout the selected measurement conditions. In contrast, TN (True Negative) is when MSD *correctly* does not detect the plasma sound when the plasma source device is inactive. Following the same pattern, FP (False Positive) is when MSD *incorrectly* detects the plasma sound when the plasma source is inactive, and FN (False Negative) is when MSD *incorrectly* does not detect the plasma sound when the plasma source is active.

Prior to the measurements, several parameters of the sound detection system must be set for each device. The same $\alpha =\left[0.59,0.99\right]$ and $b=\left[-90,-40\right]$ were used for the filter configuration for all devices along with each device’s specific $f_{f}$: $f_{f}^{\mathrm{kINPen}}=2550$ Hz, $f_{f}^{\mathrm{plasmacare}}=4042$ Hz, and $f_{f}^{\mathrm{PlasmaDerm}}=298$ Hz. For the threshold curve generation, a 12th order (j=12) polynomial regression and $\eta^{\mathrm{kINPen}}=\left[1,1.25\right]$, $\eta^{\mathrm{plasmacare}}=\left[1.42,1.775\right]$, and $\eta^{\mathrm{PlasmaDerm}}=\left[0.5,0.625\right]$ were used. Lastly, $\epsilon^{\mathrm{kINPen}}$ = 100 Hz, $\epsilon^{\mathrm{plasmacare}}$ = 250 Hz, $\epsilon^{\mathrm{PlasmaDerm}}$ = 10 Hz, $\theta^{\mathrm{kINPen}}$ = 3, $\theta^{\mathrm{plasmacare}}$ = 3, and $\theta^{\mathrm{PlasmaDerm}}$ = 25 were used for the parameters in the final detection step.

Table 2 presents the average sensitivity and specificity of the detection system tested on all three devices from all measurement conditions at each measurement distance. It is shown that the detection system is highly specific at all measurement distances for all plasma source devices. In terms of sensitivity, the measurement distance is shown to affect the detection performance. The sensitivity decreased as the measurement distance increased, which correlates with the decreasing signals from the devices. kINPen^®^ MED showed a steady high sensitivity from 5 to 20 cm and then decreased gradually in larger measurement distances, where the system could barely detect the signals from the device at 200 cm. PlasmaDerm^®^ Flex and plasma care^®^ showed a more gradual deterioration of the sensitivity, with the lowest sensitivity observed on plasma care^®^. This lower sensitivity may be due to the generated sound by plasma care^®^ being notably quieter when used with a target compared to the other devices. Nonetheless, the system is shown to deliver satisfying average sensitivity and specificity, >0.99 and 1, respectively, at a 5 cm measurement distance for all plasma source devices.

Since the sensitivity appeared to be highly affected by the measurement distance, the sensitivity was also calculated and plotted in three distinct measurement environments: quiet environment (conditions 1a and 1b), loud environment (conditions 2a and 2b), and moderate environment (conditions 3a and 3b). This was to see the behavior of the detection system with respect to the measurement distances and noises.

As shown in Figure 9a–c, respectively, for kINPen^®^ MED, plasma care^®^, and PlasmaDerm^®^ Flex, the 5 cm measurement distance persists on delivering the best sensitivity, where in most conditions, the sensitivity reached above 0.9. Intuitively, the loud environment should have affected the detection performance the most negatively, and the quiet environment should have delivered the best sensitivity at all measurement distances. However, it was not always the case in this system due to the adaptive thresholding and filtering technique implemented in this detection system. With plasma care^®^ and PlasmaDerm^®^ Flex, the loud environment did affect the detection performance negatively: 0.898 and 0.986 of sensitivity at 5 cm, respectively. The negative trend continued for the rest of measurement distances. Whereas with kINPen^®^ MED, the loud environment appeared to deliver a more robust detection (sensitivity of one) from 5 to 30 cm compared to other conditions, where the sensitivity of one could only be observed up to 20 cm. For plasma care^®^, the loud and moderate environments affected the detection performance more negatively than they affected other devices, which is again owed to the low sound signals of plasma care^®^.

Lastly, the average $t^{\det}$ for each device was calculated. The measurement script recorded how many detection instances, *D*, happened in all measurement instances, *M*. In this case, *M* is equal to 48: six measurement conditions, each with eight different measurement distances. In addition, every $t^{\det}$ for each detection instance was recorded. Based on these data, the average detection time can be calculated:(13)$${\bar{t}}^{\;}\det=\frac{1}{D}\sum_{i=0}^{D-1}t_{i}^{\det}.$$

As shown in Table 3, the system has similar average $t^{\det}$ values for all three devices, which in this case is around 0.37 s. Based on this information, the average processing time, ${\bar{t}}^{\mathrm{proc}}$, can be calculated by subtracting ${\bar{t}}^{\det}$ with $t^{\mathrm{rec}}$, ${\bar{t}}^{\mathrm{proc}}\approx 0.27s$. Although $t^{\mathrm{measure}}$ was set to 30 s for one measurement instance, the total of $t^{\det}$ for this instance may not be equal to $t^{\mathrm{measure}}$. A slight discrepancy may occur because of the system’s technical limitation. This discrepancy is calculated using:(14)$$\mathrm{Error}=\left|1-\frac{\sum_{i=0}^{D-1}t_{i}^{\det}}{\sum_{i=0}^{M-1}t_{i}^{\mathrm{measure}}}\right|.$$

The discrepancies for all plasma source devices were calculated to be less than one percent, as shown in Table 3.

### 3.2. Example Use Case of MSD

Right after starting up MSD, the user is presented with the main screen, as shown in Figure 7a, where current readings of ambient temperature and humidity as well as time and date are displayed. The main screen also consists of several virtual buttons, which trigger different functionalities, such as changing the GUI language, starting a new treatment session, browsing existing treatment reports, and turning off the system. From the main screen, the user follows a step-by-step process for documenting the plasma treatment, which is structured as follows: (1) starting a new treatment session, (2) filling in the patient’s general information using the provided virtual keyboard (see Figure 7b), (3) choosing which plasma source device will be used (see Figure 7c), and finally, (4) placing the microphone at the appropriate place and starting the logging process. During the logging process, the detection is visualized (see Figure 7d) and logged alongside the humidity and temperature measurements.

After the treatment is completed, the user stops the logging process, and then, a treatment report is generated based on the recorded data. Figure 10 shows an example of a plasma treatment report generated using MSD, which compiles the patient’s information, the plasma application duration, the average temperature and humidity throughout the treatment, and the treatment time and date. Optional additional images can be implemented such as wound images per treatment day via an external camera. The report is saved in MSD and can be viewed again later on. Additionally, the saved reports can be exported in a PDF format to an external USB drive as needed.

## 4. Conclusions and Outlook

In this paper, an early development of a mobile sensory device prototype, named MSD, is presented. The invention aims to improve the logging of treatment conditions when applying medical plasma source devices in ambient environments outside an acclimatized clinical setting. A plasma sound detection system was developed to determine the plasma treatment time, which is an important parameter for the evaluation of the treatment results. Furthermore, MSD is equipped with a temperature and humidity sensor for the logging and documentation of the ambient conditions.

Two versions of MSD were developed: standalone and an attachable module. The plasma sound detection system was validated in six measurement conditions at eight different measurement distances. The measurement results show that a high detection sensitivity (>0.99) can be achieved at a 5 cm measurement distance in all conditions for the kINPen^®^ MED and PlasmaDerm^®^ Flex devices. For the plasma care^®^ device, a high detection sensitivity was observed in most conditions except from the loud environment condition, where it only scored 0.898. Conversely, a high detection specificity was observed in all measurement conditions at all measurement distances for all plasma source devices. From this result, it can be concluded that MSD’s microphone may have to be attached to the plasma sources as close as possible to the nozzle or electrode (in case of DBD sources), and there needs to be minimal noise in the treatment environment to achieve the most optimal detection. For the microphone placement off the plasma source (e.g., on the patient’s cloth close to the wound), the correct orientation and placement of the microphone must be ensured as to have a clear and unobstructed signal acquisition.

Future development of MSD will consider adding the feature of wound image acquisition and automatic wound size measurement, which is of importance as far as wound treatment documentation is concerned. Measurements of the emitted reactive species, such as ozone (O${}_{3}$) and nitrogen dioxide (NO${}_{2}$), that are released into the ambient environment might be of interest from research and safety standpoints. Gas sensors to measure these reactive species in the ambient environment can be added to the system in a similar manner to that of DHT22. To improve the handling of the plasma source device operation, the future iteration will also consider implementing concept methods as described in [31,32]. With the integration of the mentioned features, MSD might provide a complete solution for plasma wound treatment documentation.

All in all, the first iteration of the developed MSD prototype offers a more automated, integrated, and streamlined approach of documenting important parameters in the plasma treatment, especially in the plasma medicine use cases, which may help clinicians and researchers in achieving a more complete plasma treatment documentation with ease and minimal user variability.

## Figures and Tables

**Figure 1 sensors-22-07242-f001:**
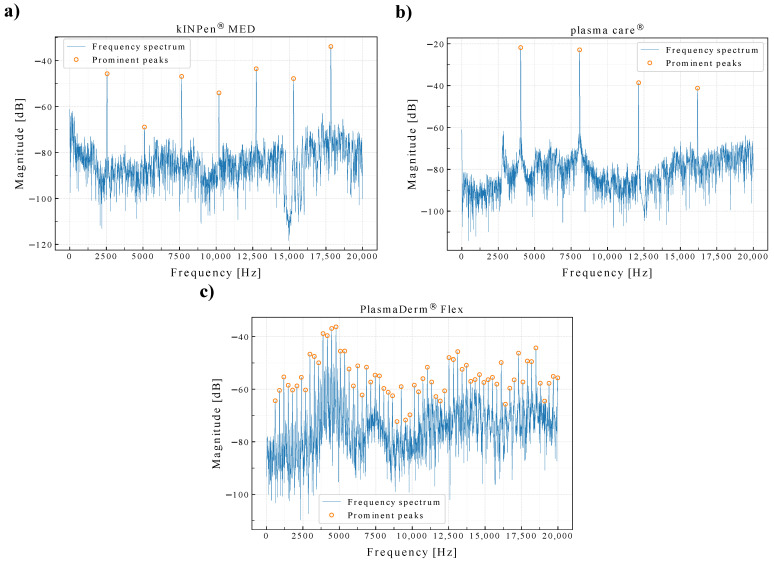
Sound frequency spectra (sampling rate = 44.1 kHz and sample size = 100 ms) and the prominent peaks of (**a**) kINPen^®^ MED, (**b**) plasma care^®^, and (**c**) PlasmaDerm^®^ Flex.

**Figure 2 sensors-22-07242-f002:**
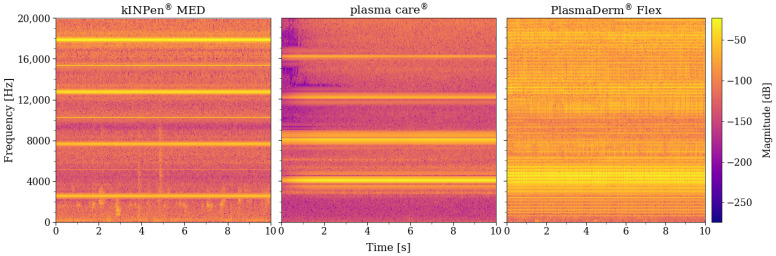
Sound spectrograms of the plasma source devices (sampling rate = 44.1 kHz, FFT size = 100 ms, observation length = 10 s).

**Figure 3 sensors-22-07242-f003:**
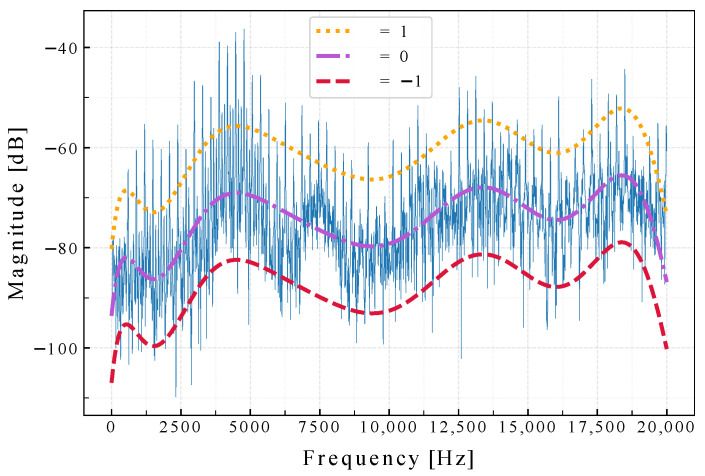
Twelfth order polynomial threshold curves of PlasmaDerm^®^ Flex sound frequency spectrum with different η values on top of the PlasmaDerm^®^ Flex sound frequency spectrum.

**Figure 4 sensors-22-07242-f004:**
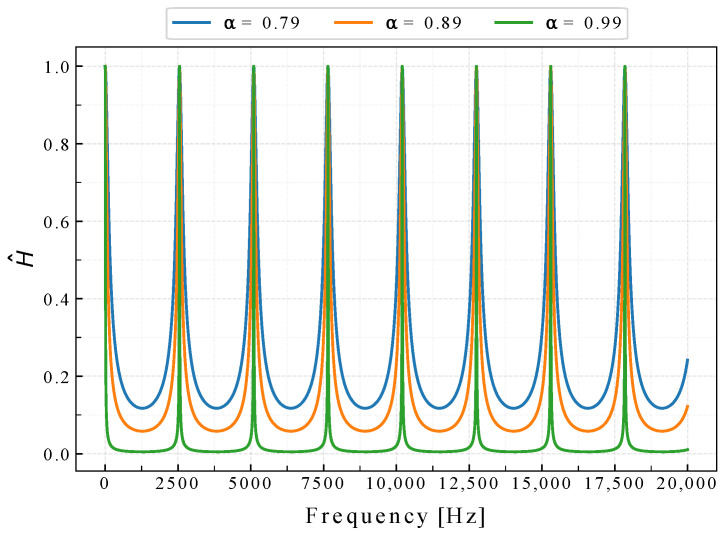
Normalized filters with different α values with $f_{f}=2550$ Hz.

**Figure 5 sensors-22-07242-f005:**
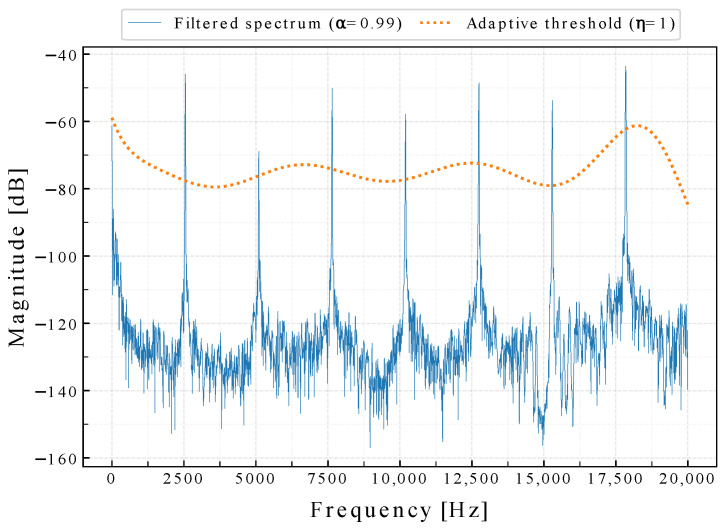
Filtered kINPen^®^ MED sound frequency spectrum, with a filter configuration: α=0.99, and $f_{f}=2550$ Hz as well as the corresponding threshold curve. Note that the threshold generation step is completed before the filtering process and uses the unfiltered frequency spectrum data.

**Figure 6 sensors-22-07242-f006:**
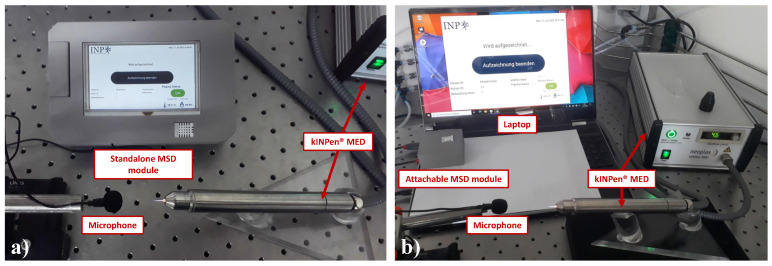
Assembled mobile sensory device (MSD) in use with the kINPen^®^ MED plasma source device. (**a**) Standalone MSD, and (**b**) MSD as an attachable module for a laptop.

**Figure 7 sensors-22-07242-f007:**
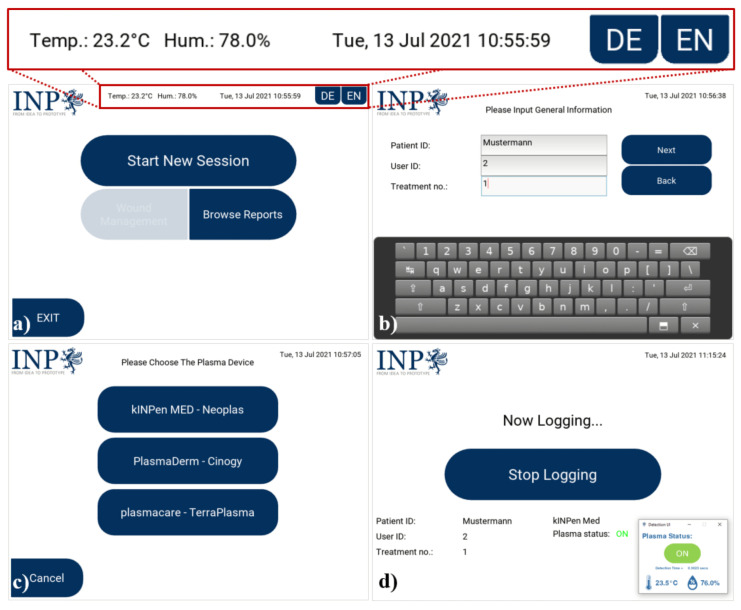
Example of the GUI screens (software version 0.3.0): (**a**) main screen, (**b**) general information input screen, (**c**) plasma source device selection screen, and (**d**) logging screen.

**Figure 8 sensors-22-07242-f008:**
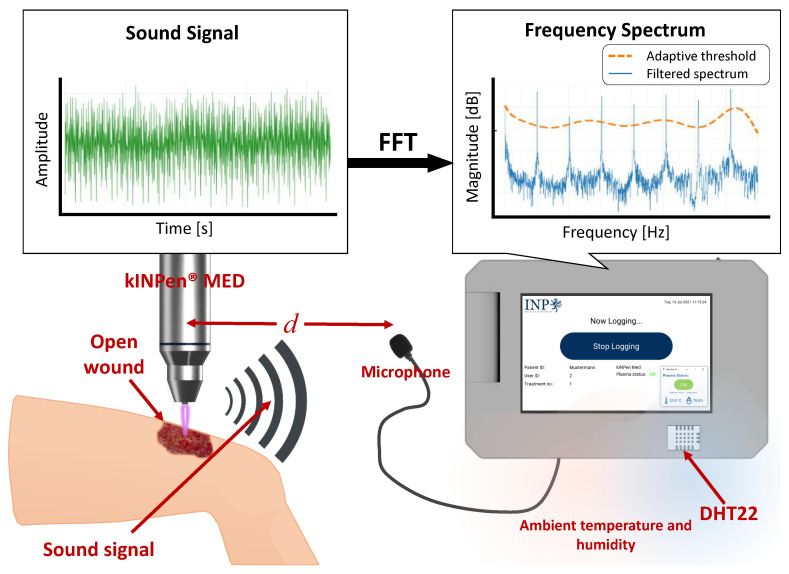
Illustration of the prototype’s working principle with the kINPen^®^ MED device.

**Figure 9 sensors-22-07242-f009:**
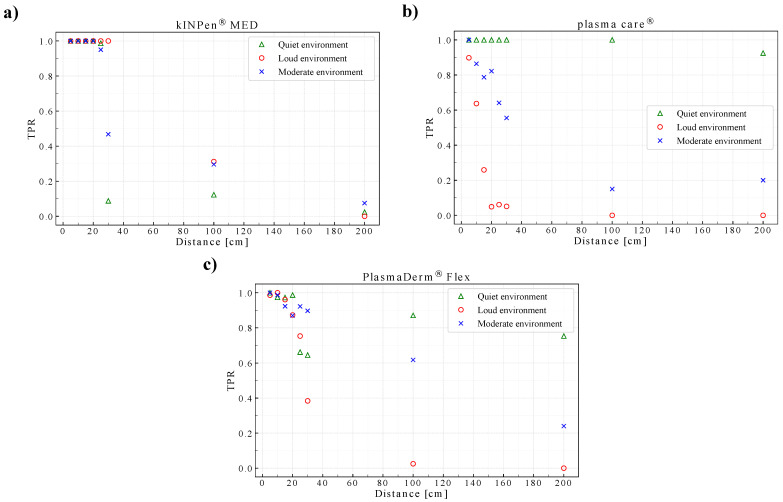
Sensitivity (TPR) of the detection system tested on the (**a**) kINPen^®^ MED, (**b**) plasma care^®^, (**c**) PlasmaDerm^®^ Flex devices.

**Figure 10 sensors-22-07242-f010:**
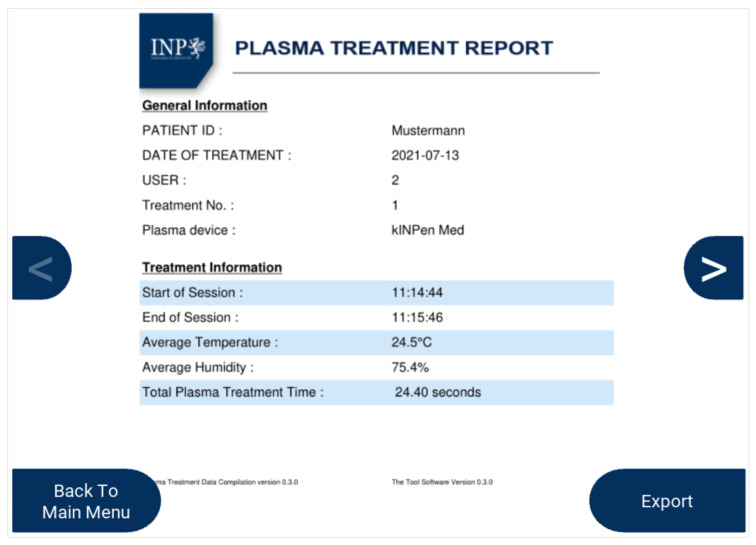
Example of the generated plasma treatment report.

**Table 1 sensors-22-07242-t001:** Fundamental and target frequencies of the devices between 0 and 20,000 Hz (audible frequency range).

Device	$f_{f}\;\left[\mathrm{Hz}\right]$	$f^{\mathrm{target}}\;\left[\mathrm{Hz}\right]$
kINPen^®^ MED	2550	$\left\{2550x\;|\;x\in N\wedge 0>x\leq \lfloor 20,000/2550\rfloor \right\}$
plasma care^®^	4040	$\left\{4040x\;|\;x\in N\wedge 0>x\leq \lfloor 20,000/4040\rfloor \right\}$
PlasmaDerm^®^	300	$\left\{300x\;|\;x\in N\wedge 0>x\leq \lfloor 20,000/300\rfloor \right\}$

**Table 2 sensors-22-07242-t002:** Average sensitivity (TPR) and specificity (TNR) of all there devices based on the data collected from the measurements in all conditions. Best scores are presented in bold.

	kINPen^®^ MED		Plasma Care^®^		PlasmaDerm^®^ Flex	
*d* [cm]	TPR	TNR	TPR	TNR	TPR	TNR
**5**	**1**	**1**	**0.966**	**1**	**0.995**	**1**
10	1	1	0.834	1	0.987	0.995
15	1	1	0.679	1	0.952	1
20	1	1	0.617	1	0.910	1
25	0.979	1	0.564	1	0.779	1
30	0.521	1	0.542	1	0.642	1
100	0.243	1	0.381	1	0.504	1
200	0.033	1	0.375	1	0.329	1

**Table 3 sensors-22-07242-t003:** Standalone MSD’s average detection time for all plasma source devices.

Device	*D*	$\Sigma \;t^{\mathrm{measure}}\left[s\right]$	$\Sigma \;t^{\det}\left[s\right]$	${\bar{t}}^{\det}\left[s\right]$	Error [%]
kINPen^®^ MED	3837	1440	1438.975	0.375	0.071
plasma care^®^	3839	1440	1445.186	0.376	0.36
PlasmaDerm^®^ Flex	3732	1440	1436.561	0.384	0.238

## Data Availability

Not applicable.

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
