# Peer review of "Development of a Mobile Sensory Device to Trace Treatment Conditions for Various Medical Plasma Source Devices"

_sensors, 2022, doi:10.3390/s22197242_

Round 1

Reviewer 1 Report

Comments on manuscript sensors-1893824 ‘Development of a mobile sensory device to trace treatment conditions for various medical plasma source devices’ by Siffa et al.

This paper reported describes the development of a mobile sensory device (referred to as MSD) that can be used during the treatment to ease the documentation of important parameters in a streamlined process. MSD was developed as a standalone Raspberry Pi-based version and attachable module version for laptops and tablets. Both versions feature a user-friendly GUI, temperature-humidity sensor, microphone, treatment report generation and export. The tool described in this paper is an interesting method to a more automated, integrated, and streamlined approach of plasma treatment documentation that can help prevent user variability. However, some concerns listed below should be take into account before publication in Sensors.

1.     The sound signals for the manuscript were recorded in a quiet confined room (Line 136), but how to exclude the interference from ambient noise in practical applications.

2.     In addition to measurement distance (Table 2), the effects of the relative position of the microphone and plasma source on signal acquisition should not be neglected in MSD applications.

3.     What is the main reason for such a significant difference in TPR detection of the three plasma sources (Table 2)? Is it possible to improve the detection sensitivity by optimizing the algorithm or the device structure?

4.     What causes the threshold curve shown in Figure 5 to not fit as well as Figure 3?

Author Response

We thank the reviewer for their constructive comments on our manuscript. We have been able to incorporate changes to reflect the suggestions provided by the reviewer. The changes are highlighted in the revised manuscript (diff-main.tex) and referenced within the corresponding response.

The following is a point-by-point response to the reviewer’s comments:

  1. Comment: The sound signals for the manuscript were recorded in a quiet confined room (Line 136), but how to exclude the interference from ambient noise in practical applications.

Response: It is true that the sound data used for the analysis were recorded in a quiet room. To address the ever-present ambient noises in practical applications, we have introduced an adaptive detection threshold and filtering mechanisms (line 176-, and line 200-). The robustness of the detection system was measured, and it shows that a proper detection in a noisy environment still can be achieved within reasonable measurement distance.

  1. Comment: In addition to measurement distance (Table 2), the effects of the relative position of the microphone and plasma source on signal acquisition should not be neglected in MSD applications.

Response: Thank you for the remark. You have raised an important point here. This will definitely have an impact on the signal acquisition when the microphone is placed off the plasma source, e.g., on the patient's cloth close to the wound. A fold on the cloth may obstruct the microphone, and the orientation of the microphone could be incorrect, i.e., the microphone does not face the sound source. We have adjusted the manuscript to incorporate this point (line 408 - 410). We would also like to mention that for the best treatment detection, we recommend attaching the microphone to the plasma source, therefore incorrect orientation and placement can be avoided.

  1. Comment: What is the main reason for such a significant difference in TPR detection of the three plasma sources (Table 2)? Is it possible to improve the detection sensitivity by optimizing the algorithm or the device structure?

Response: The sound analysis and how the plasma source devices are used suggest that the loudness of the plasma sounds affect the detection TPR. For example, the kINPen Med device has the loudest sound signal, which in turn gives the best TPR (averaged over all measurement distances). Meanwhile, the plasma care device has the lowest performing TPR, this is because when the device is used according to its standard operating procedure, it can dampen the produced sounds quite significantly. We are not completely sure yet on how to improve the device structure to address this matter. We have been entertaining the idea of using a deep learning technique for the plasma sound detection, which may perform better in a condition where the target signals are low and ambient noises are over-dominating. Nonetheless, we argue that the current state of the detection system is sufficient for the ambulant use cases.

  1. Comment: What causes the threshold curve shown in Figure 5 to not fit as well as Figure 3?

Response: Thank you for pointing this out. After another look at these two figures, we can understand that such a visual discrepancy can appear. We would like to point out that this is probably due to the fact that Figure 5 shows a filtered frequency spectrum and an adaptive threshold curve, whereas Figure 3 shows an unfiltered frequency spectrum. And please note that the threshold curve is generated using the data from the unfiltered frequency spectrum. We have extended the caption for Figure 5 to re-emphasize the threshold generation step.

Reviewer 2 Report

The work presented by the authors is well organized, described, and thoughtful in implementation. A detection device for plasma treatment in the medical setting is needed to establish parameters for clinicians to understand how much treatment is needed and the correct 'dose' for applications. The current work is a proof of concept with further work required to meet the clinical challenges.

I have two study design questions based on the approach:

1. In your introduction you envision this product being used in an ambulance setting for wound treatment. How does the 'loud environment' test compare to that of an ambulance? I completely understand that is an end-goal and not a current one, but it may interest the reader how close the loud environment is to the real ambulance setting.

2. For wound treatment, many times the actual wound is wet and could dampen the plasma sound from the plasma cure and plasmaderm devices. Was this considered? How do the researchers think this will impact their result?

Author Response

We thank the reviewer for their constructive comments and suggestions on our manuscript. We have been able to incorporate changes to reflect the suggestions provided by the reviewer. The changes are highlighted in the revised manuscript (diff-main.tex) and referenced within the corresponding response.

The following is a point-by-point response to the reviewer’s comments:

  1. Comment: In your introduction you envision this product being used in an ambulance setting for wound treatment. How does the 'loud environment' test compare to that of an ambulance? I completely understand that is an end-goal and not a current one, but it may interest the reader how close the loud environment is to the real ambulance setting.

Response: Thank you for pointing this out and for the suggestion. Here, we may have not been clear on how the ambulant setting is supposed to be in this case (plasma wound treatment). For this kind of treatment, an ambulant setting here means that a nurse would come with a plasma device to a patient’s place (residential building) and conduct the treatment there. We assume that the chosen loud noises in the manuscript would mimic the typical residential ambient noises (people talking, news on TV or radio, music, etc.). We have adjusted the text in the introduction section (as a footnote on page 3) to incorporate this suggestion, that is to make the definition of the ambulant setting clearer.

  1. Comment: For wound treatment, many times the actual wound is wet and could dampen the plasma sound from the plasma cure and plasmaderm devices. Was this considered? How do the researchers think this will impact their result?

Response: Thank you for this remark. You have a very important point here and we are happy about this input. In practice, it is not only the humidity of the wound surface but also the geometry of the wound, e.g., depth versus large area will generate a different sound intensity profile at the point of using the microphone albeit it is still not known how significant the differences are. The impact is comparable to what the other reviewer pointed out with the angle of sound acquisition towards the source of the sound. For a practical application, the TPR has to be evaluated again under real application conditions. Nonetheless, we believe that the developed detection system may still perform reasonably well under these different surface conditions provided the most optimal microphone placement (e.g., attached as close as possible to the plasma source).

Round 2

Reviewer 1 Report

 This manuscript offers a integrated approach in the plasma medicine use cases, and this work need to be promoted.